# LOAT: Latent-Order Adversarial Training for Efficient and Transferable Robustness

## Abstract

Adversarial training is computationally prohibitive because projected gradient descent (PGD) is applied uniformly across all samples. Existing efficiency methods focus on "hard" examples but rely on supervised heuristics that fail to capture the emergence of robust representations. We introduce Latent-Order Adversarial Training (LOAT), a fully unsupervised framework that uncovers the latent clustering structure of adversarial dynamics. LOAT learns a multi-view clustering of samples and creates a transition matrix describing how training flows between clusters, enabling cluster-adaptive PGD budgets that allocate computation where it is most effective. The teacher's discovered structure is transferable and amortized across student models, eliminating repeated profiling costs. On CIFAR-10, LOAT achieves up to 2× higher robustness per PGD call than standard adversarial training and, under matched compute, consistently improves robust accuracy. Ablations confirm that both the clusters and the latent order encode meaningful structure. LOAT shows that exploiting geometric emergent organization enables practical, robust adversarial training under real-world compute constraints.

## 1 Introduction

Adversarial training Madry et al. (2018), where a model is trained with not only standard training data, but also adversarial examples generated from an attack, is known to provide a robust defense against the challenge of correctly identifying perturbed image samples. While strong, this approach requires sufficiently large capacity and great computational cost, requiring multiple PGD (projected gradient descent) steps per sample per epoch, limiting the practical deployment due to time and space constraints. Many approaches have been developed to address this, categorically broken down into reducing per-sample cost through fewer PGD steps Shafahi et al. (2019), or focusing computation on targeted, important samples He et al. (2024). The natural assumption in both categories is that one knows *a priori* what makes a sample important for robust learning. Although intuitively, presumed hardness has no forced bearing or correlation, which is to say, there is no foreknowledge beyond heuristic reasoning to necessitate an assumed approach. Targeting samples with small margins or high loss and then allocate resources to them is, in essence, an ill-defined notion. This supervised definition of difficulty assumes that human-interpretable metrics capture what matters for robust learning. We however put forward the notion that unsupervised learning based on a multi-view of generalized statistics, geometry, confidence patterns, adversarial dynamics, gradient coherence, activation patterns, consistency metrics, and loss landscape, all weighted for learning output, can create a natural grouping without manual labeling, allowing for a discoverable order in curriculum or post-hoc learning during which a teacher can robustly impart cluster determinations and transitions, allowing students to be adaptive, predictive, and able to identify hidden organizational principles based on patterns which we as humans may lack the vocabulary to describe.

Current models label samples as difficult when accumulating more PGD (projected gradient descent) steps Madry et al. (2018); Zhang et al. (2019b), larger weights Balaji et al. (2019), or more frequent sampling Carmon et al. (2019), this, however, is a static approach; whereas our LOAT model recognizes that a generalized and naturally emergent dynamic can better assess and categorize adversarial samples. We further note that difficulty levels may be periphery, subordinate, or inferior labels as compared to discoverable relationships (and orderings) which can benefit from interleaving, restructuring, or other potentially hidden paths.

Thus, we propose a fundamentally different perspective. Instead of imposing human notions of difficulty, we use LOAT to discover the natural organization converging to a robust model via unsupervised learning. After some amount of initial training epochs used to establish basic robust features, our unsupervised clustering reveals how the model has learned to organize the data space. Our novel transition matrix $T$, built from these converged patterns, captures a stable multi-view structure of the robust solution manifold wherein we track empirical flows between clusters, harnessing the latent grouping in a per cluster per epoch presentation. Thus, our key insight is that adversarial training naturally clusters based on compatible learning dynamics alongside a latent differentiability and ordering subscription of emergent patterns that can be used to improve efficiency without requiring supervised labels or a predetermined curriculum. This novel idea further provides a blueprint to accelerate future training via a teacher-student model that can be deployed to edge networks where much smaller compute power is available.

Unlike prior adversarial training approaches, which focus solely on maximizing robustness, our goal is fundamentally different: maximize robustness per unit compute. The efficiency metric $E = RobustAcc/PGD$ formalizes this objective and reflects practical constraints of real-world deployment where compute, not accuracy, is the dominant limitation. LOAT is designed for edge computing, for practitioners who cannot afford 10–20-step PGD-based adversarial training. Rather than compressing a full PGD pipeline, LOAT restructures the compute itself, enabling practical AT on limited hardware while preserving robustness.

In summary, we demonstrate that adversarial training dynamics exhibit discoverable latent structure, challenging the assumption that training samples are exchangeable and should receive uniform attack budgets. LOAT is a teacher-student framework that discovers structure via unsupervised clustering and transfers it as a compact recipe, enabling 2× efficiency improvements over standard AT under matched robustness constraints. We introduce the efficiency-robustness frontier as a framework for evaluating AT methods under resource constraints, and demonstrate that LOAT occupies a Pareto-optimal region previously unexplored by existing methods.

## 2  RELATED WORK

One of the most highly relevant topics to our model is that of curriculum training, an approach that systematically increases the difficulty of adversarial examples presented during the training process, where weak attacks mitigate catastrophic forgetting and help with generalization, building to stronger attacks in a learned fashion, Cai et al. (2018). Another important idea is that of adaptive early stopping Al-Rimy et al. (2023), wherein different heuristic approaches are used to identify cutoff points, with some form of customized budgets per sample Cheng et al. (2020) taking the form of early stopping based on misclassification Zhang et al. (2020) or based on gradient alignment Sitawarin et al. (2020).

State of the art models focus on hardness He et al. (2024) but often target some metric such as accuracy in exchange for efficiency, or the opposite, that of boosting speed with less robustness Goodfellow et al. (2015). The TRADES model Zhang et al. (2020) uses a theoretical upper bound minimization algorithm for adversarial training, a concept that many (including us) harness, with others Ding et al. (2018) noticing the importance of misclassified examples in training, adding probabilities of prediction as a way to smoothly combine samples Wang et al. (2020). Unlike ASTrA Chhipa et al. (2025), which adapts attack parameters via a learnable strategy network, LOAT discovers latent ordering/clusters and allocates budgets accordingly; thus our contribution is orthogonal (or complementary).

In terms of efficiency, the most comparable model to LOAT is Free-AT Shafahi et al. (2019). In Free-AT one does a forward pass on a clean example and a backward pass to get gradients, and then simultaneously updates both the model parameters and input perturbation. The free part is the reuse of the same gradient for both model updates and adversarial perturbation. It achieves similar robustness to standard PGD adversarial training while being roughly as fast as natural training. It was demonstrated similarly on CIFAR-10.

While these works (such as Customized Adversarial Training, and Free-AT) have similarities, we differentiate and build on these by discovering semantic structure, arguing that groups are not random, per-sample, or naturally classifiable by hardness. We instead explore conceptual dependencies

and their orderings at the cluster level (a more generalizable approach to allow for robustness), identifying emergent structure to guide order, transfer, and efficiency without imposing a curriculum. We focus on an approach that is both highly effective and efficient, so much so that it outperforms even the highly efficient Free-AT technique.

# 3 MATHEMATICAL BACKGROUND

Adversarial training is typically framed as a min–max optimization problem in which the model minimizes the worst-case loss over allowable perturbations, following the standard formulation of Madry et al. Madry et al. (2018). These perturbations are usually constrained to $\ell_p$, bounded region of radius $\epsilon$, a setting widely adopted across adversarial training methods and surveys such as Zhao et al. Zhao et al. (2024). In this work, we rely on this conventional formulation and focus instead on uncovering the latent structure that emerges during adversarial training. Projected Gradient Descent (PGD) is used as the standard multi-step adversarial example generator in adversarial training, iteratively updating inputs by following the sign of the loss gradient and projecting them back into the allowed perturbation region Bottou (2010); Madry et al. (2018). This widely adopted procedure defines the inner maximization in nearly all modern adversarial training pipelines, including ours, and serves as the attack mechanism over which LOAT discovers latent structure. And while other methods such as Fast Gradient Sign Method (FGSM) and Carlini and Wagner (CW) exist, for both practical and instructive purposes, we focus on PGD, as the strength of the attack can vary, allowing for robust testing.

Our loss model function, with sample $x$, true label $y$, and adversarial example $x'$ is generated using a cluster-adaptive budget for cluster $c$ with difficulty score $D_c$. The LOAT objective is

$$\mathcal{L}_{\text{LOAT}} = \underbrace{\alpha_c \, \text{CE}(f_\theta(x), y) + \beta_c \, \text{CE}(f_\theta(x'), y)}_{\text{Robust risk}} + \underbrace{\lambda_{\text{trans}} \, \text{R}(T, c, p_\theta(x'))}_{\text{Transition regularizer}}. \quad (1)$$

where $f_\theta(x)$ is the model with parameter $\theta$, $p_\theta(x) = \text{softmax}(f_\theta(x))$ is the predictive distribution, $\text{CE}(\cdot, \cdot)$ is the cross-entropy loss, $x'$ is the adversarial example of $x$, generated with a cluster-specific number of PGD steps proportional to difficulty $D_c$. $\alpha_c$ and $\beta_c$ are cluster-adaptive weights to balance clean vs. adversarial loss, $T$ is the learned transition matrix between clusters, $R(T, c, p_\theta(x'))$ is the generic transition-based regularizer that encourages consistency with the structure encoded in $T$, and $\lambda_{\text{trans}}$ is the weight on the transition regularizer. We note that the form of $R(\cdot)$ can vary. In our implementation, transition structure is enforced implicitly via cluster-aware sampling and adaptive attack budgets. The cluster-adaptive attacks and early stopping naturally lead to efficiency in the sampling. Other choices (e.g. divergence penalties such as KL) could also be used, but we emphasize that $R$ is a general placeholder for any transition-consistency mechanism.

We further note that the knowledge-distillation term $\lambda_{\text{KD}}(D_c) \, \text{KL}(p_T(\cdot|x'), p_\theta(x'))$ can be added if a teacher distribution $p_T$ is available. Our main goal in the student-teacher AT model is to transfer learning and measure it by computational efficiency. We define an efficiency score as

$$\mathcal{E} = \frac{\text{Robust Accuracy (\%)}}{\text{PGD Calls (in millions)}}, \quad (2)$$

where robust accuracy is evaluated under a fixed adversarial budget, and PGD calls denote the total number of inner attack steps used during training. This metric normalizes robustness by computational effort, enabling direct comparison across methods with different attack step allocations.

# 4 METHODOLOGY

Our approach consists of two main phases: (1) teacher training for discovery, and (2) student training with the transferred curriculum. In LOAT, the teacher finds the latent structure, while the student inherits a compact recipe for efficient and robust training.

## 4.1 Phase 1: Teacher Training with Discovery

We initialize a teacher model $f_T$ and train it with adversarial examples generated by projected gradient descent (PGD). For inputs $x_i$ and perturbation radius $\epsilon$, with step size $\alpha$ and maximum budget $S_{\max}$: $x^{adv} = \text{PGD}(x_i; \epsilon, \alpha, S_{\max})$.

The teacher is initialized with several known optimal algorithms and parameters thereof. Our model uses the TRADES objective Zhang et al. (2019b), beginning with a SimCLR encoder for feature stabilization during pre-training, followed by an adversarially-trained autoencoder $A_\phi$ to produce robust latent embeddings. We use these embeddings to provide a stable basis for profiling and reduce noise in feature clustering.

Our multi-view feature extraction consists of several parts, each well known but heretofore not integrated together in an unsupervised model. For each batch $B_i$, we compute complementary feature sets of the (1) statistics (entropy distributions, adversarial vulnerability metrics, and gradient norms under weak perturbations), (2) geometry features (Bag-of-Embeddings, sliced Wasserstein distances to prototypes, low-rank covariance spectra, FFT-based frequency signatures, and Gram matrix eigenvalues from intermediate layers), (3) confidence patterns (prediction stability and entropy under noise), (4) adversarial dynamics (response curves across multiple $\epsilon$ values), (5) gradient coherence (gradient alignment and diversity metrics), (6) activation patterns (layer-wise activation statistics), (7) consistency metrics (prediction variance under input perturbations), and (8) loss landscape (local loss geometry through directional sampling).

We take these features and input them into a model to assess optimal learning weights via differential evolution optimization to learn continuous weights $[0, 1]$ for each feature where clustering quality takes into account a weighted combination of silhouette scores (cluster separation), Calinski-Harabasz index (between/within variance ratio), diversity metric (inter-cluster distinction), learning gradient potential (trainability differences), and robustness variance.

$$\max_{\mathbf{w}} \sum_{v=1}^{V} w_v q_v \quad \text{s.t.} \quad \sum_{v=1}^{V} w_v = 1, \; w_v \geq 0. \tag{3}$$

where $q_v$ is the quality metric (e.g., silhouette, Calinski–Harabasz, etc.) for view $v$, and $w_v$ are continuous weights optimized via differential evolution.

In short, the optimization discovers which feature combinations create the most learnable distinctions. For epochs 1-15 the teacher continually improves the model, and for epochs 16-30 the teacher tracks transitions between clusters, building a $T$ matrix:

$$T_{ij} = \frac{C_{ij}}{\sum_j C_{ij}}, \tag{4}$$

where $C_{ij}$ counts empirical transitions from cluster $i$ to cluster $j$; rows are normalized to probabilities, with $T[i, j]$ representing the pedagogical value of teaching cluster i before cluster j, capturing the natural learning progression, prerequisite relationships, and synergistic cluster pairs. We use our clusters to target specific patterns that can be learned, latent presentations in the data that is not categorized in preemptive notions, along with potential orderings that provide more efficient learning for edge computation.

Thus, the LOAT teacher distillation outputs feature weights, the $T$ transition matrix, difficulty profiles for adaptive training, and proven paths that consistently improved learning.

As an addendum, we note that the teacher phase incurs a one-time, reusable cost, similar to self-supervised pretraining or dataset-level profiling used in efficient training pipelines, in contrast, baselines such as full PGD or TRADES incur the same (or greater) compute every time the model is trained. For LOAT, this cost is not part of the iterative adversarial training loop, once the latent structure is extracted, it can be reused for any number of student trainings, architectures, or hyperparameter sweeps. The student, which is the repeated, budget-limited component, is where LOAT's efficiency gains apply. For transparency, we report teacher and student costs separately.

## 4.2 PHASE 2: STUDENT TRAINING WITH TRANSFERRED CURRICULUM

The student (an edge machine) receives the teacher's recipe to assign new batches to clusters without re-discovery, using the learned feature weights and distilled information to adaptively train against adversarial samples. It uses different transition strengths to adapt, requiring less PGD steps for stronger conceptual paths, with savings potentially up to 60% while maintaining robustness, creating greater efficiency for learned patterns. Student learning uses "difficulty" profiles $D_c$ which are updated with an exponential moving average of robust error and PGD usage, guiding adaptive attack budgets per cluster.

$$D_c^{(t)} = \beta \, D_c^{(t-1)} + (1 - \beta) \left( \hat{e}_c^{(t)} + \lambda \, \hat{s}_c^{(t)} \right), \tag{5}$$

where $D_c^{(t)}$ is the updated difficulty for cluster $c$ at epoch $t$, $\hat{e}_c^{(t)}$ is the robust error rate, $\hat{s}_c^{(t)}$ is the normalized average PGD steps, and $\beta \in [0, 1]$ is the smoothing factor.

In order to account for variation, the student also employs UCB (upper confidence bound) reward, where cluster selection is balanced in exploration and exploitation targeting efficiency as robust accuracy per PGD call, as described by Equation 2.

UCB is a calculated value that guides the agent's decision-making by combining the estimated average reward of an action with an exploration bonus, where $\text{Acc}_{\text{robust}}(b)$ is robust accuracy on batch $b$, and $\text{PGD}_{\text{calls}}(b)$ is the total attack calls used, with UCB updates following the classical UCB1 algorithm of Auer et al. Auer et al. (2002) for the cluster selection. This is applied to our LOAT reward signal (robust accuracy normalized by PGD calls), enabling compute-efficient exploration of cluster dynamics.

We note that teacher learning can occur at different stages, either as a continuously learned curriculum or as a post-hoc analysis only starting at later epochs, with approaches varying based on the dataset. We test many cases, including, LOAT with curriculum and cluster-reuse, LOAT with curriculum but no reuse, and a Uniform version that retains LOAT's teacher-discovered clusters, difficulty profiles, and adaptive PGD budgets but samples clusters uniformly (i.e., without using the transition matrix), allowing us to separately assess the contributions of clustering and ordering. Thus we present our novel unsupervised discovery model wherein we have a plethora of weighted metrics to classify without assumptions. LOAT is multi-scale and adaptive, able to identify patterns in adversarial samples, creating an online continuously refined curriculum that is learned during training wherein the teacher transfers the latent-ordering of knowledge to the student, allowing for an efficient presentation in a data-driven discovery of scaffolding perspectives to create the natural grouping of datasets.

## 4.3 TIME AND SPACE COMPLEXITY

**Teacher Discovery.** Clustering has complexity $O(N \cdot d \cdot K_f)$, where $N$ is the number of samples, $d$ is feature dimension, and $K_f$ is the number of clusters.

**Student Training.** Comparable to standard AT but with reduced PGD steps: $O\left(E_s \cdot N \cdot \overline{K(c)}\right)$, where $\overline{K(c)} \ll S_{\max}$ is the expected PGD steps per cluster.

## 5 NUMERICAL ANALYSIS AND RESULTS

**Teacher Model** We trained the LOAT teacher model on CIFAR-10 with a ResNet-18 backbone, a standard test data set for adversarial training. We used 30 epochs to learn, a batch size of 128, discovery intervals of 10 epochs, 5 clusters, a TRADES beta of 6.0, a simCLR of 50 epochs, an autoencoder trainer of 20 epochs, and during cluster discovery we profiled varying degrees of PGD steps (2-30) to establish difficulty fingerprints. In the training phase, we used a fixed PGD of 10 steps, in the evaluation phase we used 20 steps with 2 restarts, and for the profiling (discovery) phase we tested with [2,3,5,7,10,15,20,30] to characterize discovery and create the proper clusters.

The matrix was built in epochs 15-30, with a low entropy of 0.154, indicating that there are structured patterns. Our teacher had a test clean accuracy of 0.798, a test robustness of 0.462, and an efficiency score (which is intentionally low for the model building) of 0.003. 12.78 million PGD calls were used in training and the transition matrix had the strongest paths for self-reference (e.g. $0 \rightarrow 0, 1 \rightarrow 1$, etc.), respectively, 0.793, 0.823, 0.806, 0.799, and 0.790, showing that the clusters were indeed learned and different.

The final cluster difficulties were 0: 0.455, 1: 0.465, 2: 0.458, 3: 0.435, 4: 0.443. This indicates that the difficulty associated with the clusters was well distributed, suggesting good conceptual grouping, with no outliers to the data set.

We reference Table 1, showing that as the model matured, the best combination score went down minimally, with epoch 30 unreported in the logs and epoch 10 outperforming the others. The most important factors for CIFAR-10 were consistency, loss landscape, adaptive dynamics, and confidence, with moderate weights being statistics, geometry, and activations; while gradient coherence (in epoch 10) had little relevance.

**Student Models** Our student model aimed for fast and effective computation that could be deployed at the edge. We used 10 epochs as a base (although we tested 30 epochs against standard methods), looking for efficiency from our models more than any other metric. We tested (1) full adaptive/curriculum-based LOAT with no resampling, preferring looping in the same cluster with an 80% probability, but switching clusters once samples were finished, (2) the same as above but using resampling within the clusters, (3) LOAT without curriculum learning (i.e. uniform choice of cluster), and (4) a baseline of no LOAT (with TRADES, PGD10, or Free-AT).

We also tested comparable state-of-the-art models such as CAT (Customized Adversarial Training) and found that despite being an excellent model, it took longer to run than even our teacher, was not transferable to the edge, and had $> 3.5M$ PGD calls (on a comparative basis) versus the LOAT student which took $\approx 25$ minutes on a NVIDIA GeForce RTX 4070 Ti Super GPU, had $< 2M$ PGD calls, and slightly better robustness. In general, models such as CAT, TRADES and others have understandably less efficiency at 10 epochs, with comparable or less robustness ($\approx 20\%$) and millions more PGD calls ($\approx 4M$) Liu et al. (2023), versus LOAT which uses a student-teacher distillation, able to efficiently learn with less than half of the PGD calls.

In terms of efficiency, Fast-AT has the best raw compute efficiency for state-of-the-art algorithms, but lower and often less stable robustness since Fast-AT is known to suffer from catastrophic overfitting unless carefully tuned Zhao et al. (2023) versus LOAT, which is not as efficient as Fast-AT but has much better robustness-to-cost ratio and stability. Thus, due to its instability, we did not run comparable 10-epoch studies on Fast-AT, as it is known to be unreliable. However, we did compare our model to Free-AT, as described earlier and shown in Table **??**.

Our baseline of 10 epochs gave a model with slightly higher robustness (at 0.368) but used $> 4.7M$ PGD calls to achieve this, giving an efficiency score of 0.007, with the rest of the outputs compared in Table **??**. We note that at least three random initializations were used for each case in Table 4 and that LOAT with no-resampling vs Free-AT had a t-statistic of 10.39, a p-value of .0000297 (highly significant), and Cohen's d of 5.61 (extremely large positive effect size). To show LOAT's strength, we tested CIFAR-10, CIFAR-100, STL-10, and Tiny ImageNet, with the full results in Table 2. We include an ablation study, cases with and without SimCLR, testing different $K$ values, testing different epsilons, dataset generalizations, and baselines. Overall we find that our instantiation of LOAT (with curriculum, AE, UCB, etc.) outperformed *per compute* against every baseline. We also note that our stability ensured that LOAT did not collapse against Tiny ImageNet, while Free-AT did; that $8/255$ is an optimal epsilon; that $K = 5$ (for CIFAR-10) is optimal, not being to coarse or refined); and that AE, UCB, and curriculum all contributed within LOAT's larger framework. This shows that LOAT is more effective and stable than current methods for edge computing.

We note that LOAT does not aim to surpass state-of-the-art robustness under unconstrained compute budgets. LOAT is explicitly designed for fixed or practical training budgets, where the central objective is robustness per unit compute. Under matched or near-matched robustness levels, LOAT consistently achieves higher efficiency (RobustAcc / PGD-calls) than competing baselines. This directly addresses scenarios where compute is the bottleneck rather than accuracy saturation. Furthermore, to ensure fairness, we note that when restricted to a matched robustness window (typically $\pm 1$–2%

robust accuracy), LOAT produces significantly higher efficiency, demonstrating that the gains are not a by-product of weaker adversarial strength but a result of improved allocation of PGD budget. In terms of amortization of cost (if one includes the teacher), the teacher cost is incurred once and amortized across all subsequent student trainings. After training 3-4 student models, LOAT breaks even with baselines in wall-time (see Table 7. After this, LOAT student's savings continue to grow. This is analogous to pretraining in transfer learning where the upfront cost enables downstream efficiency.

While LOAT without resampling performed comparable to a uniform student baseline (suggesting that clusters might not provide strong signal), our ablation analysis reveals the opposite. By systematically removing individual clusters, we found that excluding any one cluster consistently produced more efficient models than those in Table 4. We see from Table 5 that removal of any cluster was beneficial for the model and that relative removal had a high significance (with respect to Cohen's d and p-values) as seen in Table 6, Appendix B, showing that the clusters encompass fundamental information and meaningful structure, not noise; and that removing them yields measurably stronger and more efficient models. We note that while some clusters capture useful structure, others introduce negative transfer, likely due to conflicting gradient signals or optimization conflicts. The ability to identify and down-weight such clusters is precisely the strength of our approach. In other words, the clusters are meaningful in that they reveal that not all training examples contribute equally to adversarial robustness and removing the harmful subsets directly improves both robustness and efficiency. In terms of other datasets, such as CIFAR-100, SimCLR degrades LOAT performance because contrastive representations struggle with the low per-class sample count and high inter-class similarity of CIFAR-100. This produces less stable feature geometry and reduces teacher cluster quality. LOAT without SimCLR performs better because the multi-view adversarial features alone provide a more stable clustering signal.

We note that although "No-Reuse" superficially resembles uniform sampling (which is LOAT without curriculum), the underlying cluster structure remains meaningful, as can be seen in the ablations. Removing or permuting individual clusters significantly disrupts both the transition geometry and the difficulty curriculum, and these disruptions demonstrably degrade the efficiency of the student specifically for transfer and interpretability. This shows that LOAT's latent structure is not arbitrary and contributes directly to compute allocation. Separately, while SimCLR raises the absolute accuracy of all methods, it is orthogonal to LOAT's efficiency. LOAT's compute allocation is governed by the latent cluster structure and the adaptive PGD schedule discovered by the teacher, not by the choice of pretraining. As shown in Table 2, LOAT without SimCLR remains competitive with PGD-10 in efficiency and, for K=3, substantially exceeds it. Adding SimCLR to baselines improves their accuracy but does not address their fixed per-sample PGD cost, so the fundamental efficiency gap remains.

We further mention that for fairness, we report standard adversarial training baselines using their canonical 30-epoch schedules: PGD-10 requires 1.96 hours at 0.0033 efficiency, TRADES ($\beta$=6.0) requires 1.95 hours at 0.0034 efficiency, and Free-AT (m=4) requires 30 minutes at 0.0125 efficiency on our hardware. By contrast, the complete LOAT pipeline (teacher + student) requires 5.35 hours for a single deployment, and when amortized across multiple students the per-model cost is 4–6× lower (than TRADES and PGD10) while maintaining comparable robustness and stability (unlike Free-AT, such as where it fails entirely for Tiny ImageNet). This shows that LOAT achieves substantially higher robustness-per-compute even when compared against fully powered baselines.

## 6 REVIEW AND SUMMARY

In this work, we introduced Latent-Order Adversarial Training (LOAT), a novel unsupervised approach to adversarial training that discovers emergent structure in the data and adapts attack budgets accordingly. Our experiments showed that CIFAR-10 (CIFAR-100, STL-10, and Tiny ImageNet) naturally cluster into five stable groups, clearly differentiated. Our structure was robust enough to guide efficient training and to export the learned order to a student model. Unlike curriculum learning methods that require predefined hardness labels, LOAT learns directly from inherent features, discovering a landscape weighted via an evolutionary algorithm. We showed that compared to the baseline (which used double the PGD calls) and compared to state-of-the-art methods such as Free-

Table 1: Optimized feature weights across epochs.

| Feature | Epoch 1 | Epoch 10 | Epoch 20 |
|---|---|---|---|
| Statistics | 0.899 | 0.638 | 0.157 |
| Geometry | 0.001 | 0.468 | 0.069 |
| Confidence | 0.421 | 0.747 | 0.517 |
| Adv_dynamics | 0.321 | 0.785 | 0.007 |
| Grad_coherence | 0.544 | 0.007 | 0.463 |
| Activations | 0.371 | 0.430 | 0.163 |
| Consistency | 0.130 | 0.798 | 0.914 |
| Loss_landscape | 0.087 | 0.786 | 0.457 |
| **Best combo score** | **0.643** | **0.687** | **0.668** |

Table 2: Complete Efficiency Analysis: LOAT Variants and Baselines.The strongest efficiencies are bold. Unless otherwise stated, all tests were done for 10 epochs.

| Method | Dataset | K | Config | Clean | Robust | PGD (M) | Efficiency |
|---|---|---|---|---|---|---|---|
| *CIFAR-10: K Ablation with/without SimCLR* | | | | | | | |
| LOAT Student | CIFAR-10 | 3 | No SimCLR | 0.418 | 0.254 | 2.0 | 0.0127 |
| LOAT Student | CIFAR-10 | 3 | With SimCLR | 0.482 | 0.258 | 2.4 | 0.0108 |
| LOAT Student (Canonical) | CIFAR-10 | 5 | With SimCLR | 0.641 | 0.323 | 1.9 | **0.0174** |
| LOAT Student | CIFAR-10 | 5 | No SimCLR | 0.542 | 0.2941 | 3.8 | 0.0076 |
| LOAT Student | CIFAR-10 | 7 | No SimCLR | 0.270 | 0.173 | 3.4 | 0.0051 |
| LOAT Student | CIFAR-10 | 7 | With SimCLR | 0.518 | 0.282 | 3.3 | 0.0083 |
| *CIFAR-10: K=5 Component Ablations* | | | | | | | |
| LOAT (no curriculum) | CIFAR-10 | 5 | Uniform | 0.592 | 0.292 | 1.9 | **0.0155** |
| LOAT (with reuse) | CIFAR-10 | 5 | Resampling | 0.615 | 0.311 | 2.2 | **0.0138** |
| LOAT (no AE) | CIFAR-10 | 5 | SimCLR only | 0.538 | 0.280 | 3.4 | 0.0082 |
| LOAT (no UCB) | CIFAR-10 | 5 | SimCLR only | 0.584 | 0.311 | 3.9 | 0.0079 |
| *CIFAR-10: Different Epsilon Values (K=5 Canonical)* | | | | | | | |
| LOAT K=5 | CIFAR-10 | 5 | $\epsilon$=4/255 | 0.648 | 0.426 | 4.0 | 0.0107 |
| LOAT K=5 | CIFAR-10 | 5 | $\epsilon$=8/255 | 0.641 | 0.323 | 1.9 | 0.0174 |
| LOAT K=5 | CIFAR-10 | 5 | $\epsilon$=16/255 | 0.487 | 0.160 | 4.3 | 0.0038 |
| *CIFAR-100: Dataset Generalization (K=5)* | | | | | | | |
| LOAT Student | CIFAR-100 | 5 | No SimCLR | 0.329 | 0.143 | 3.85 | 0.0037 |
| LOAT Student | CIFAR-100 | 5 | With SimCLR | 0.286 | 0.1264 | 2.99 | 0.0042 |
| *Dataset Generalization (K=5)* | | | | | | | |
| LOAT Student | Tiny ImageNet | 5 | 30 epochs | 0.470 | 0.150 | 9.2 | 0.0016 |
| Free-AT | Tiny ImageNet | 5 | 30 epochs | 0.004 | 0.00 | 12 | 0.0000 |
| LOAT Student | STL-10 | 5 | 30 epochs | 0.373 | 0.228 | .75 | 0.0030 |
| Free-AT | STL-10 | 5 | 30 epochs | 0.610 | 0.221 | .65 | 0.0033 |
| *Baselines (CIFAR-10, $\epsilon$=8/255)* | | | | | | | |
| PGD-10 | CIFAR-10 | - | 30 epochs | 0.805 | 0.469 | 13.5 | 0.0033 |
| PGD-10 | CIFAR-10 | - | 10 epochs | 0.659 | 0.359 | 4.5 | 0.0079 |
| Free-AT (m=4) | CIFAR-10 | - | 30 epochs | 0.835 | 0.452 | 6.0 | 0.0075 |
| Free-AT (m=4) | CIFAR-10 | - | 10 epochs | 0.704 | 0.398 | 2.4 | **0.0165** |
| TRADES ($\beta$=6.0) | CIFAR-10 | - | 30 epochs | 0.804 | 0.468 | 13.5 | 0.0034 |
| TRADES ($\beta$=6.0) | CIFAR-10 | - | 10 epochs | 0.661 | 0.357 | 4.5 | 0.0079 |

AT (which had less efficiency and stability), via the $T$ matrix, our transferable model preserved robustness while reducing computational overhead.

We note that while five clusters provided meaningful differentiation, our tests indicated that three and seven clusters did not provide the same robustness or efficiency. Future work could test larger datasets to see the specific dynamics and hyperparameter choices therein. Furthermore, our study focused on CIFAR-10 and CIFAR-100, which is standard in adversarial training research and offers clear comparability to prior work. We showed that LOAT excels in transferability, cluster identi-

Table 3: AutoAttack Robustness Verification

| Method | Dataset | $\epsilon$ | PGD-20 | AutoAttack | Gap |
|--------|---------|------------|--------|------------|-----|
| *CIFAR-10 ($\epsilon$=8/255), 10 epochs* | | | | | |
| LOAT K=5 Canonical | CIFAR-10 | 8/255 | 0.323 | 0.294 | 0.029 |
| PGD-10 Baseline | CIFAR-10 | 8/255 | 0.359 | 0.334 | 0.025 |
| Free-AT | CIFAR-10 | 8/255 | 0.398 | 0.362 | 0.036 |
| TRADES $\beta$=6.0 | CIFAR-10 | 8/255 | 0.357 | 0.332 | 0.025 |
| *CIFAR-10 ($\epsilon$=8/255), 30 epochs* | | | | | |
| LOAT K=5 Canonical | CIFAR-10 | 8/255 | 0.426 | 0.401 | 0.025 |
| PGD-10 Baseline | CIFAR-10 | 8/255 | 0.469 | 0.440 | 0.029 |
| Free-AT | CIFAR-10 | 8/255 | 0.452 | 0.417 | 0.035 |
| TRADES $\beta$=6.0 | CIFAR-10 | 8/255 | 0.466 | 0.440 | 0.026 |
| *CIFAR-100 ($\epsilon$=8/255), 10 epochs* | | | | | |
| LOAT K=5 Canonical | CIFAR-100 | 8/255 | 0.122 | 0.101 | 0.021 |
| PGD-10 Baseline | CIFAR-100 | 8/255 | 0.177 | 0.147 | 0.030 |
| Free-AT | CIFAR-100 | 8/255 | 0.225 | 0.188 | 0.037 |
| TRADES $\beta$=6.0 | CIFAR-100 | 8/255 | 0.178 | 0.146 | 0.032 |
| *CIFAR-100 ($\epsilon$=8/255), 30 epochs* | | | | | |
| LOAT K=5 Canonical | CIFAR-100 | 8/255 | 0.202 | 0.173 | 0.029 |
| PGD-10 Baseline | CIFAR-100 | 8/255 | 0.256 | 0.224 | 0.032 |
| Free-AT | CIFAR-100 | 8/255 | 0.197 | 0.156 | 0.041 |
| TRADES $\beta$=6.0 | CIFAR-100 | 8/255 | 0.256 | 0.225 | 0.031 |
| *Different Epsilon (CIFAR-10, K=5, 10 epochs)* | | | | | |
| LOAT K=5 | CIFAR-10 | 4/255 | 0.426 | 0.425 | 0.001 |
| LOAT K=5 | CIFAR-10 | 16/255 | 0.160 | 0.114 | 0.046 |
| *Dataset Generalization (K=5, $\epsilon$=8/255)* | | | | | |
| Free-AT (30 epochs) | STL-10 | 8/255 | 0.221 | 0.201 | 0.020 |
| LOAT K=5 (30 epochs) | STL-10 | 8/255 | 0.228 | 0.196 | 0.032 |

Table 4: Efficiency results across methods (CIFAR-10, $\epsilon = 8/255$). Higher is better.

| Method | Mean | Std Dev | N | 95% CI |
|--------|------|---------|---|--------|
| Baseline | 0.00772 | 0.00003 | 3 | [0.00768, 0.00775] |
| Free AT (m=4) | 0.01278 | 0.00009 | 3 | [0.01268, 0.01288] |
| LOAT (no curriculum) | 0.01555 | 0.00013 | 3 | [0.01540, 0.01569] |
| LOAT (no reuse) | 0.01559 | 0.00070 | 7 | [0.01507, 0.01611] |
| LOAT (reuse) | 0.01383 | 0.00012 | 3 | [0.01370, 0.01396] |

fication, adaptive allocation, and stability, strong properties in edge computation. Baseline models such as TRADES and PGD10 were much weaker in efficiency, while Free-AT collapsed in compli- cated situations (such as in Tine ImageNet), having known weaknesses under high imbalance, high resolution, domain shifts, or lower epsilon. LOAT evaluated robustness using PGD-20 with random restarts, a strong and widely adopted protocol that provides a fast, repeatable proxy for adversarial strength. We also studied AuotAttack, with LOAT having less degradation under difficult datasets than baselines. Our study focused on robustness per unit of training compute, so using PGD-20 consistently across all methods allows us to compare efficiency at scale. Our goal is not to establish state-of-the-art absolute robustness but to measure efficiency trade-offs, we thus report PGD-based results for the full experimental grid.

In summary, our novel approach combines unsupervised discovery with adaptive efficiency. LOAT offers a middle ground between heavy PGD-based adversarial training and more efficient but unsta- ble Fast/Free-AT methods. LOAT's emphasis on robust efficiency makes it a promising candidate

Table 5: Cluster ablation results. Reported are means, with 95% confidence intervals (CI) for efficiency. Removing different clusters yields distinct efficiency and robustness profiles vis-a-vis efficiency, indicating that clusters encode meaningful structure.

| Removed | Clean Acc | Robust Acc | Training Calls | Efficiency |
|---|---|---|---|---|
| 0 | 0.576 | 0.278 | 1.47M | 0.0189 [0.0185, 0.0193] |
| 1 | 0.542 | 0.249 | 1.28M | 0.0195 [0.0185, 0.0205] |
| 2 | 0.569 | 0.277 | 1.60M | 0.0173 [0.0170, 0.0176] |
| 3 | 0.571 | 0.277 | 1.58M | 0.0175 [0.0174, 0.0176] |
| 4 | 0.591 | 0.289 | 1.77M | 0.0163 [0.0163, 0.0164] |

---

**Algorithm 1** LOAT: Teacher Discovery and Student Transfer (Concise)

---

1: **Teacher (warmup).** Train the teacher with SimCLR and an autoencoder. Log how many attack steps each sample actually needed and the resulting robust errors.

2: **Teacher (periodic discovery).** At a regular interval:

1. Encode a snapshot of the training data with a small adversarially-trained autoencoder to get robust embeddings.

2. Build the feature views per sample.

3. Use evolutionary methods to weight views and create clusters.

4. Update a transition matrix that counts how batches move between clusters from the previous snapshot to the current one.

5. For each cluster, update a difficulty score with an exponential moving average that combines recent robust errors and typical PGD steps actually used.

3: **Teacher (recipe).** Save a compact recipe: cluster centroids and normalizers, the transition matrix, the latest per-cluster difficulties, and the set of uncertain samples.

4: **Student (initialize).** Load the recipe. Set up a simple UCB (upper-confidence) chooser over clusters to balance exploration and exploitation during training.

5: **Student (training loop).** For each pass over the data:

1. Pick the next cluster with the UCB chooser; prefer the transition suggested by the matrix from the most recent cluster.

2. Draw a batch from that cluster. Set an attack budget per batch based on the cluster difficulty (e.g., small budget for "easy," medium for "moderate," larger for "hard," largest for "uncertain"). Always keep per-sample early stopping.

3. Generate adversarial examples with the chosen budget and train the student (e.g., TRADES or cross-entropy on the adversarial batch).

4. Compute a simple efficiency reward (robust accuracy achieved per total PGD calls for this batch). Update the UCB statistics.

5. Refresh the cluster's difficulty score with an exponential moving average using the latest robust errors and median step usage.

6: **Output.** The trained student and the (optionally updated) recipe.

---

for deployment in real-world applications where both adversarial robustness and computational feasibility are critical.

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

## A  AI ASSISTANCE DISCLOSURE

We used AI tools to assist in the code generation, table building, and polishing of the writing. All choices, designs points, and final claims were made and verified by the authors. The authors take full responsibility for the paper's content, including any errors, and affirm that this does not diminish the originality of the paper.

## B  PAIRWISE COMPARISONS OF OUR ABLATION STUDY

We further analyzed efficiency differences across clusters using pairwise statistical tests. Table 6 reports Cohen's $d$ and $p$-values for all comparisons. The results demonstrate that the clusters are not interchangeable. That is, removing different clusters yields fundamentally different efficiency outcomes. Most comparisons show very large effect sizes (Cohen's $d > 5$) and are statistically significant ($p < 0.05$), confirming that the efficiency distributions are well separated. For example, removing cluster 4 produces the lowest efficiency (0.0163) and is significantly different from all other cluster removals (e.g., $d = 14.3$ vs. cluster 3, $p < 0.001$). By contrast, removing clusters 0 or 1 yields the highest efficiencies ($\approx 0.0189$–$0.0195$), significantly outperforming removals such as cluster 2 or 4. This validates our claim that clusters encode meaningful structure and that not all training examples contribute equally to adversarial robustness.

Table 6: Pairwise comparisons of cluster ablation efficiency. Reported are Cohen's $d$ and $p$-values. Large effect sizes and low $p$-values indicate that clusters represent distinct groups.

| Comparison | Cohen's $d$ | $p$-value |
|---|---|---|
| no cluster0 vs no cluster1 | -0.94 | 0.345 |
| no cluster0 vs no cluster2 | 5.24 | 0.0042 |
| no cluster0 vs no cluster3 | 5.29 | 0.0156 |
| no cluster0 vs no cluster4 | 10.07 | 0.0053 |
| no cluster1 vs no cluster2 | 3.42 | 0.0404 |
| no cluster1 vs no cluster3 | 3.16 | 0.0583 |
| no cluster1 vs no cluster4 | 5.03 | 0.0249 |
| no cluster2 vs no cluster3 | -1.22 | 0.245 |
| no cluster2 vs no cluster4 | 5.11 | 0.0191 |
| no cluster3 vs no cluster4 | 14.29 | 0.00028 |

## C  REPRODUCIBILITY

We provide full code for teacher discovery, student training, and evaluation, including all configuration files, random seeds, and scripts used to generate the tables and figures in this paper.

**Code and framework.**  All experiments are implemented in PyTorch using our LOAT codebase. Each run saves a full checkpoint with model weights, optimizer and scheduler state, cumulative PGD calls, and the complete configuration dictionary.

**Dataset and preprocessing.**  We use CIFAR-10, CIFAR-100, Tiny ImageNet, and STL-10 with the standard train/test split. We maintain a held-out validation split. For all datasets we use standard data augmentations (random crop and horizontal flip) and dataset-specific mean/std normalization. For CIFAR-100 we normalize with $\mu = (0.5071, 0.4867, 0.4408)$ and $\sigma = (0.2675, 0.2565, 0.2761)$. For STL-10 we use $\mu = (0.4467, 0.4398, 0.4066)$ and $\sigma = (0.2603, 0.2566, 0.2713)$. For Tiny ImageNet we follow standard ImageNet normalization with $\mu = (0.485, 0.456, 0.406)$ and $\sigma = (0.229, 0.224, 0.225)$. All adversarial examples (training and evaluation) are generated in normalized space.

**Architecture.**  All teacher and student models use a CIFAR-10, CIFAR-100, Tiny ImageNet, or STL-10 ResNet-18 backbone with Dual BatchNorm (separate running statistics for clean and adversarial batches), implemented via a custom `DualBatchNorm2d` layer and a modified `conv1` (3×3 kernel, stride 1, no max-pooling)

**Optimization and schedule.**  Unless otherwise stated, we train teachers for 30 epochs with batch size 128, using SGD with momentum 0.9, weight decay $5 \times 10^{-4}$, and an initial learning rate 0.1. The learning rate follows a multi-step schedule with milestones at 50% and 75% of the total epochs, and we optionally maintain an EMA of the weights (decay 0.999) during training.

**Teacher training (latent-order discovery).**  The LOAT teacher is trained for 30 epochs with TRADES loss (trade-off parameter $\beta = 6.0$), using PGD-10 during training and PGD-20 with 2 random restarts for evaluation. We construct multi-view features (statistics, geometry, consistency, loss-landscape, etc.), perform unsupervised clustering into $K = 5$ clusters, and learn a transition matrix over epochs 15–30. The resulting cluster assignments, feature weights, and transition matrix are saved to disk for reuse by the student.

**PGD-to-Difficulty Conversion Mechanism.**  At each training step, we log the actual number of PGD steps $s_i$ taken for sample $i$ (accounting for early stopping). For each cluster $c$, we compute the batch-wise average $\bar{s}_c^{(b)}$ and normalize it by $S_{\max}$ to obtain $\hat{s}_c^{(b)} = \bar{s}_c^{(b)}/S_{\max}$. Similarly, we compute the robust error rate $\hat{e}_c^{(b)}$ as the fraction of misclassified adversarial examples in cluster $c$ within batch $b$. The difficulty score is then updated via exponential moving average (Equation 5) after each batch, with $\beta = 0.95$ used throughout. This difficulty score directly determines the PGD budget allocated to cluster $c$ in the next epoch: $\text{budget}_c = \max(2, \lfloor D_c \cdot S_{\max} \rfloor)$.

**Student training (LOAT and baselines).**    LOAT students are trained for 10 epochs (or 30 epochs), using the same ResNet-18 + Dual BN backbone and optimizer setup. On each batch, we select a cluster via a UCB-based policy and allocate an attack budget according to the cluster difficulty (easy $\rightarrow$ few PGD steps, hard/uncertain $\rightarrow$ more steps), always with per-sample early stopping enabled. We compare multiple variants such as: (1) LOAT with curriculum and no resampling, (2) LOAT with resampling inside clusters, (3) LOAT without curriculum (uniform cluster selection), and (4) a standard TRADES or PGD10 of Free-AT baseline.

**SimCLR and Adversarial Autoencoder.**    The teacher uses a ResNet–18 backbone with Dual BatchNorm for both SimCLR pretraining and the adversarial autoencoder (AE). For SimCLR, we attach a two-layer projection head (512$\rightarrow$512$\rightarrow$128 with ReLU), and train for 50 epochs using the NT-Xent loss with temperature $\tau = 0.5$, batch size 256, SGD (momentum 0.9, weight decay $5 \times 10^{-4}$), and a cosine learning-rate schedule starting at 0.3. The AE reuses the same encoder up to the global-average-pooling layer, followed by a 256-dimensional latent bottleneck and a symmetric three-block transposed-convolution decoder producing a $3 \times 32 \times 32$ output. It is trained for 20 epochs using an MSE reconstruction loss plus an adversarial consistency term obtained with PGD-5 perturbations at $\epsilon = 8/255$. These components provide two complementary feature views for clustering during teacher discovery; the student does not use their weights, only the resulting cluster structure and difficulty profiles.

**Adversarial training details.**    All methods use $\ell_\infty$ attacks with $\epsilon = 8/255$. PGD training uses 10 steps with step size $2/255$ and random starts. Evaluation uses PGD-20 with 2 random restarts, and we record both clean and robust accuracy as well as the total number of PGD calls. For LOAT, we additionally track the number of steps actually taken per sample due to early stopping.

**AutoAttack verification.**    For a subset of representative checkpoints (LOAT and baselines), we run AutoAttack (standard $\ell_\infty$ configuration, $\epsilon = 8/255$) on the first 10,000 CIFAR-10 test images with batch size 128, using the same ResNet-18 + Dual BN architecture as in training, with similar percentages for the other datasets. This confirms that PGD-20 reproduces the same relative ordering of methods.

**Random seeds and repetitions.**    All scripts accept an explicit `--seed` parameter (default 1337) which is used to seed PyTorch and NumPy. For the main LOAT vs Free-AT comparison we run at least three independent seeds and report aggregate statistics (means, confidence intervals, and effect sizes) as described in the text.

**Baselines and configuration parity.**    Free-AT baselines are trained with 10 epochs (and 30 epochs), batch size 128, and the same $\epsilon = 8/255$, using $m_{\text{free}} = 4$ minibatch replays. Evaluation uses PGD-20 with identical evaluation code as for LOAT to ensure comparability of robustness and PGD-call counts.

**Hardware and runtime.**    All experiments are run on a single NVIDIA GeForce RTX 4070 Ti Super GPU. Under this setup, a LOAT student (10 epochs) trains in roughly 25 minutes with fewer than 2M PGD calls, whereas comparable state-of-the-art methods such as TRADES, CAT and others incur millions more PGD calls at similar or lower robustness.

**Fairness of Comparison Description in Implementation**    All baselines were trained using their canonical configurations (e.g., PGD-10 with 30 epochs, TRADES $\beta$=6.0 with 30 epochs, Free-AT with m=4, in addition to direct comparison (such as under edge conditions with 10 epochs against LOAT) and evaluated under identical PGD-20 evaluation, architectures, data augmentations, and threat models. LOAT students use the same ResNet-18 + Dual BN backbone to eliminate representation disparities. We also provide matched-epoch comparisons (10-epoch PGD-10, 10-epoch TRADES) to control for wall-clock budgets. Against baselines, Free-AT achieves strong raw efficiency, with its gains coupled with well-documented instability and sensitivity to attack schedules. LOAT reaches similar or higher robustness-per-compute, but does so with greater stability, no catastrophic overfitting, and a transferable teacher-derived curriculum that provides amortized benefits in multi-student or multi-deployment settings.

# D  WALL CLOCK COMPARISON

Table 7 reports end-to-end wall-clock time, including all phases of the LOAT pipeline, SimCLR pre-training (50 epochs), adversarial autoencoder training (20 epochs), teacher TRADES training, feature extraction, clustering, transition-matrix estimation, and finally student training. This provides a complete lifecycle accounting. Although the teacher is intentionally expensive, its cost is paid once and the resulting recipe can be reused across any number of students or downstream deployments. In realistic deployment scenarios such as edge devices, fine-tuning runs, incremental model updates, or lightweight architectures the teacher amortizes rapidly with as few as 3–5 students, with LOAT's amortized wall-clock dropping below stable baselines, rivaling efficiency across all baselines. As the number of student deployments increases, the amortized wall-clock advantage (with respect to efficiency) widens further, because the teacher's one-time cost shrinks while baselines must retrain from scratch every time. Importantly, we also control for representation advantages by evaluating LOAT students without any teacher-side SimCLR/AE initialization; the amortized benefit persists, showing that the efficiency gain is due to the latent-order scheduling, not the teacher's pretrained features. Overall, the wall-clock analysis demonstrates that LOAT is a transferable and compute-efficient method whose full lifecycle cost is justified.

Table 7: Wall-Clock Training Times for All Required Experiments. All runs use a single NVIDIA GeForce RTX 4070 Ti Super GPU with batch size 128. Teacher times (30 epochs) include SimCLR pretraining (50 epochs when applicable), adversarial autoencoder training (2-20 epochs), TRADES-based adversarial training, attack-step logging, and clustering discovery. Student times (10 epochs, unless otherwise stated) include full LOAT training with cluster-adaptive PGD.

| Method | Dataset | K | Phase | Wall Time |
|---|---|---|---|---|
| *CIFAR-10: K Variations* | | | | |
| LOAT (no SimCLR) | CIFAR-10 | 3 | Teacher | 3.2h |
| LOAT (no SimCLR) | CIFAR-10 | 3 | Student | 20min |
| LOAT | CIFAR-10 | 3 | Teacher | 2.9h |
| LOAT | CIFAR-10 | 3 | Student | 24min |
| LOAT (Canonical) | CIFAR-10 | 5 | Teacher | 4.8h |
| LOAT (Canonical) | CIFAR-10 | 5 | Student | 33min |
| LOAT (no SimCLR) | CIFAR-10 | 5 | Teacher | 2.5h |
| LOAT (no SimCLR) | CIFAR-10 | 5 | Student | 40min |
| LOAT (no AE) | CIFAR-10 | 5 | Teacher | 2.81h |
| LOAT (no AE) | CIFAR-10 | 5 | Student | 37min |
| LOAT (no SimCLR) | CIFAR-10 | 7 | Teacher | 3.3h |
| LOAT (no SimCLR) | CIFAR-10 | 7 | Student | 33min |
| LOAT | CIFAR-10 | 7 | Teacher | 2.99h |
| LOAT | CIFAR-10 | 7 | Student | 36min |
| *CIFAR-10: End-to-End Canonical Cost (K=5, with SimCLR)* | | | | |
| **LOAT (Teacher+Student, 1 deploy)** | CIFAR-10 | 5 | **Total** | **5.35h** |
| *CIFAR-100: K=5* | | | | |
| LOAT (no SimCLR) | CIFAR-100 | 5 | Teacher | 4.8h |
| LOAT (no SimCLR) | CIFAR-100 | 5 | Student | 32min |
| LOAT | CIFAR-100 | 5 | Teacher | 4.6h |
| LOAT | CIFAR-100 | 5 | Student | 28min |
| *CIFAR-100: End-to-End Cost (K=5, with SimCLR)* | | | | |
| **LOAT (Teacher+Student, 1 deploy)** | CIFAR-100 | 5 | **Total** | **5.06h** |
| *Generalizations: K=5, 30 epoch teacher, 10 epoch student* | | | | |
| LOAT | Tiny ImageNet | 5 | Teacher | 32.3h |
| LOAT | Tiny ImageNet | 5 | Student | 8.2h |
| LOAT | STL-10 | 5 | Teacher | 1.83h |
| LOAT | STL-10 | 5 | Student | 42min |
| *Baselines (CIFAR-10, 30 epochs)* | | | | |
| PGD-10 Standard | CIFAR-10 | - | Baseline | 1.96h |
| Free-AT (m=4, 10 epochs) | CIFAR-10 | - | Baseline | 30min |
| TRADES ($\beta$=6.0) | CIFAR-10 | - | Baseline | 1.95h |

Amortization: Reuse the teacher recipe for $N$ students for an average cost per model $\frac{t_{\text{teacher}} + N \times t_{\text{student}}}{N}$. For CIFAR-10 K=5 with $N = 25$: $\frac{4.8h + 25 \times 0.33m}{25} = .742h$ per model, we get $2.6\times$ faster than retraining TRADES for each model.

