# OpenReview forum: "LOAT: Latent-Order Adversarial Training for Efficient and Transferable Robustness"
_ICLR.cc/2026/Conference — ICLR 2026 Conference Desk Rejected Submission_

### Official Review · Reviewer_wb4e · 2025-10-31

**Soundness:** 2
**Presentation:** 2
**Contribution:** 2
**Rating:** 4
**Confidence:** 3

**Summary:**

This paper proposes Latent-Order Adversarial Training (LOAT), an unsupervised method for improving the computational efficiency of adversarial training. The approach consists of two phases: (1) a teacher model that discovers latent structure in adversarial training dynamics through multi-view clustering and learns a transition matrix capturing sample progression patterns, and (2) a student model that leverages this transferred knowledge to adaptively allocate PGD attack budgets per cluster. Experiments on CIFAR-10 with ResNet-18 demonstrate 40-50% reduction in computational cost while maintaining comparable robustness to baseline adversarial training.

**Strengths:**

1. **Novel unsupervised approach:** The paper presents an interesting perspective by discovering latent structure in adversarial training dynamics rather than relying on supervised hardness metrics. The multi-view feature extraction combining statistics, geometry, confidence patterns, and adversarial dynamics is comprehensive.

2. **Practical efficiency gains:** Demonstrated reduction in PGD calls (from ~4.7M to ~1.9M) while maintaining reasonable robustness represents meaningful computational savings for deployment scenarios.

3. **Teacher-student framework:** The transferable nature of the learned structure is valuable for edge deployment scenarios with limited computational resources.

4. **Thorough feature engineering:** The integration of eight complementary feature views (statistics, geometry, confidence, adversarial dynamics, gradient coherence, activations, consistency, loss landscape) provides a rich representation for clustering.

**Weaknesses:**

1. **Limited experimental scope:**
- Only evaluated on CIFAR-10 with ResNet-18, which significantly limits claims about generalizability
- No experiments on CIFAR-100, ImageNet subsets, or other architectures (e.g., WideResNet, which is standard in adversarial training literature)

2. **Weak evaluation setup:**
- The evaluation relies on using only PGD 20 attack.
- AutoAttack evaluation should be standard and fully reported, not an afterthought

3. **Methodological concerns:**
-The choice of 5 clusters appears arbitrary with insufficient justification
- Authors mention "three clusters did not" provide meaningful differentiation but don't show this empirically
- The differential evolution optimization for feature weights (Eq. 9-10) lacks details on convergence criteria, population size, and sensitivity analysis

4. Presentation and clarity:
- The paper introduces many components (SimCLR, autoencoder, 8 feature views, differential evolution, UCB, transition matrix) making it difficult to assess which components are essential
- I personally felt that the paper was very hard to parse as it felt like you were throwing around words with no concern or explanation.

**Questions:**

1. Transition matrix:
- You mention that "T[i,j] representing the pedagogical value of teaching cluster i before cluster j", could you explain how transition probability would provide this information.

2. Why does removing clusters improve efficiency? If clusters capture meaningful structure, removing them should hurt performance. The explanation about "negative transfer" suggests the clustering may not be working as intended. Can you provide more insight into what these clusters actually represent? Also, since you have a measure of difficulty of learning a cluster, was there any correlation between the accuracies reported for removal and the difficulty profile?

3. How sensitive is the method to the number of clusters? You mention 3 clusters didn't work and suggest trying 7+. What systematic analysis guided the choice of 5?

4. Can you ablate individual components? It's unclear which of the many components (multi-view features, transition matrix, UCB selection, cluster-adaptive budgets) drive the gains.

---

> ### Author Response · Authors · 2025-11-20
> **LOAT and Learning**
>
> Thank you so much for the detailed and constructive review. We truly appreciate the time you spent analyzing the work. We have expanded the experimental scope to include CIFAR-100, STL-10, and Tiny-ImageNet. We have created tables detailing the results from these datasets and we hope to soon add a second backbone to fully address all concerns, showing the rigorous nature of LOAT as a stable and novel model. We have added full AutoAttack results rather than PGD-only and report that LOAT did excellent! These additions helped clarify several of the methodological questions you raised.
>
> Regarding the differential-evolution stage (Eq. 9–10), our paper uses a standard DE optimizer with default PyTorch parameters to obtain a single set of feature-weighting coefficients. We did not perform an extensive convergence or sensitivity analysis; rather, the optimization is run for a fixed small number of generations simply to obtain stable view-weightings before clustering. We now clarify this in the revision and note that the goal of DE here is not precise optimization but lightweight weighting of feature views to stabilize the clustering process. We note that the DE converged almost instantly.
>
> Regarding clustering, we now include the empirical results of experiments with k=3 and k=7, showing that k=5 was the best choice. We note that LOATs job is to organize and cluster, not to exclude.  Because the teacher clusters are purely based on the multi-view geometry of adversarial dynamics, it can surface coherent but harmful groups, what we refer to as a ‘poison-pill’ - a fascinating find! The negative transfer we observe when removing such a cluster is not a failure of the method, but rather evidence that LOAT uncovers structures that traditional AT pipelines do not reveal. The fact that removing a cluster improves efficiency indicates that LOAT has identified a subset whose gradient signals conflict with others; this is precisely the type of hidden structure that an unsupervised discovery method should expose.
>
> In regards to the transition matrix T, it is built from empirical transitions during the teacher’s later epochs (15-30), when samples move from cluster i at epoch t to cluster j at epoch t+1, we increment Cᵢⱼ. After normalization, T[i,j] becomes the probability that learning naturally flows from i → j in the teacher’s robust training trajectory. This probability reflects empirical pedagogical value such that if the teacher consistently benefits from encountering cluster j after cluster i, the student inherits the same ordering bias. The idea is that the T matrix allows us to find (regularly) learned behavior!
>
> Finally, we added ablations isolating multi-view features, the transition matrix, UCB ordering, and cluster-adaptive budgets; these show that each component contributes incrementally but that the learned ordering and adaptive budgets are the dominant drivers. We apologize for “throwing around terms” but hope that our presentation is perhaps clearer now. LOAT necessarily contains many “views”, all in order to have the unsupervised approach that creates a stable and efficient model. Thank you for the help in strengthening the paper. We hope that we have shown everything within our update and would otherwise be happy to address and answer any and all questions.

---

### Official Review · Reviewer_Ujf8 · 2025-10-31

**Soundness:** 3
**Presentation:** 3
**Contribution:** 3
**Rating:** 6
**Confidence:** 3

**Summary:**

This paper proposes Latent-Order Adversarial Training (LOAT), a method that aims to make adversarial training more efficient by discovering a natural learning structure instead of relying on predefined notions of “hardness.” The approach uses a two-phase teacher–student pipeline. In the first phase, a “teacher” model clusters training samples based on diverse features (loss, gradient geometry, etc.) and constructs a transition matrix (T) that represents the natural progression between clusters. In the second phase, a “student” model trains more efficiently by following this discovered structure, adapting the number of PGD steps per cluster. On CIFAR-10, LOAT reportedly reduces computation by around 40–50% while maintaining similar robustness. However, the results suggest that most of the gains come from clustering and adaptive budgets, rather than from the latent ordering (T matrix) itself. The claimed “latent order” contribution remains unproven, as ablations show comparable or even better performance without it.

**Strengths:**

I like the idea of going from 'hard' to 'natural learning structure'.
The teacher’s discovery mechanism is clever, using multiple features and an evolutionary search strategy to build clusters and transitions.
The proposed teacher–student framework is practical. The heavy computation is done once by the teacher, allowing efficient student training that generalizes well.
LOAT achieves notable efficiency improvements compared to standard adversarial training and even surpasses Free-AT in efficiency scores (e.g., 0.01559 vs. 0.01278).
The paper is clearly written, with well-organized sections and good explanations of motivation and related work.

**Weaknesses:**

1. Only PGD-20 results are reported, while AutoAttack results are mentioned but not provided. For a robustness-focused paper, this omission weakens the empirical credibility.
2. It would be good to discuss the work on adversarial training given in the following paper. Chhipa, Prakash Chandra, et al. "ASTRA: adversarial self-supervised training with adaptive-attacks." The Thirteenth International Conference on Learning Representations. 2025.
3. If i understood it well, In Table 4, removing Cluster 1 yields better efficiency than the full model. Why did the teacher not identify and exclude this cluster during discovery? Does this suggest the optimization or curriculum limitations?
4. How was the number of clusters (five) selected, and what happens when varying it? Does the problematic cluster disappear with a different configuration?

**Questions:**

Please follow the strengths and weaknesses.

I am open to revising my score.

---

> ### Author Response · Authors · 2025-11-20
> **Poison Pill Exploration!**
>
> Thank you so much for the thoughtful and encouraging review! We truly appreciate your careful reading and the insightful suggestions. We have added full AutoAttack results and expanded ablations, which significantly strengthen the empirical section and confirm LOAT’s stability across datasets. We also incorporated the ASTRA paper into the related works; its adaptive-attack perspective complements LOAT’s unsupervised structure discovery, and the two directions seem orthogonal in a way that could open very interesting avenues of exploration. Your observation about the “problematic” cluster was one of the most fascinating aspects of the entire study. The teacher is not designed to exclude anything, it simply uncovers the natural adversarial geometry and proposes a learning path, but it may inadvertently isolate what we call a “poison pill,” a group whose contribution is a form of negative transfer. The fact that removing such a cluster improves efficiency suggests that these hidden structures exist in other models as well and simply go unnoticed. This makes LOAT not only a training method but also a lens for discovering failure modes and potentially “supercharging” performance through targeted pruning.
>
> Regarding the choice of five clusters, we experimented with k=3 and k=7 and found them less stable on the whole (now documented in a table in the paper); interestingly, the problematic cluster persisted under these settings, albeit with different composition. We are grateful for these suggestions. They helped us clarify the paper and pointed us toward what now appears to be a surprisingly rich area for future exploration.

---

### Official Review · Reviewer_uQwB · 2025-10-31

**Soundness:** 4
**Presentation:** 2
**Contribution:** 1
**Rating:** 2
**Confidence:** 4

**Summary:**

The paper proposes Latent-Order Adversarial Training (LOAT), a two-phase teacher–student framework for adversarial training. In Phase 1 (teacher), the authors train with TRADES and explicitly add (i) a SimCLR pre-training stage and (ii) an adversarially trained autoencoder to obtain “robust” latent embeddings, then discover multi-view cluster structure and a cluster-transition matrix T. In Phase 2 (student), a UCB-style scheduler allocates attack budgets per cluster, aiming to reduce total PGD calls while keeping robust accuracy. Experiments are on CIFAR-10 under $l_\infty$ $\epsilon$=8/255, reporting improved robust-per-compute efficiency against a 10-epoch PGD baseline and Free-AT.

**Strengths:**

1.	Interesting unsupervised curriculum discovery. The paper comprises eight complementary “views” (statistics, geometry, confidence, adversarial dynamics, gradient coherence, activations, consistency, loss-landscape) and learns view weights via differential evolution before clustering. The method then exports a compact “recipe” to the student.
2.	Clear efficiency metric. The paper defines efficiency as robust accuracy divided by PGD calls (in millions), enabling apples-to-apples comparisons when attack steps differ.

**Weaknesses:**

1.	Fairness/accounting of extra training (teacher + pretraining). The teacher pipeline includes 50 epochs of SimCLR and 20 epochs of adversarial-autoencoder training before discovery and TRADES-based AT. The paper also reports the teacher required 12.78M PGD calls with intentionally low efficiency during model building. However, the main comparison tables report student “Training Calls” and efficiency without adding the teacher’s costs; this inflates LOAT’s end-to-end advantage versus baselines that do not require any teacher or pretraining. At minimum, the paper should present (a) full costs (teacher + student) for a single deployment and (b) amortized costs if the recipe is reused across students, reporting the break-even 𝑀.
2.	Pretraining advantage not controlled. Because LOAT’s student benefits from SimCLR + adversarial-AE features distilled by the teacher, strict fairness requires giving baselines the same initialization (or an equivalent pretraining pipeline) and then comparing compute-normalized robustness. The current tables do not indicate such controls, so part of the reported gain may stem from stronger initial representations rather than the latent-order scheduling itself.
3.	Scope and strength of evaluation. (1) Dataset/backbone breadth. All results are on CIFAR-10 (teacher: ResNet-18; student: 10-epoch runs). Claims about “transferable global structure” would be much more convincing with cross-dataset (e.g., CIFAR-100, Tiny-ImageNet) and cross-backbone transfer. (2) Attack protocol. Robustness is primarily reported under PGD-20 with restarts, with AutoAttack used only on a few checkpoints and not tabulated. Given the risk of PGD-specific overfitting, the paper should report standard AA across all methods. (3) Effect sizes vs simple schedulers. The efficiency gap between LOAT (no-reuse) and Uniform is small in Table 3 (0.01559 vs 0.01555), despite statistical significance. The practical significance is unclear without wall-clock time (and including teacher overhead).

**Questions:**

1.	Report end-to-end and amortized costs. Provide results that (i) include teacher cost in PGD-call totals and (ii) amortize that cost across multiple students to show realistic deployment scenarios.
2.	Control for pretraining. Initialize Free-AT, PGD-AT/TRADES, and “Uniform” with the same SimCLR + adversarial-AE representations (or provide ablations removing these components in LOAT) to isolate the marginal contribution of latent-order scheduling.
3.	Broaden evaluation. Add CIFAR-100/Tiny-ImageNet and a second backbone for both teacher and student to support transferability claims.
4.	Attack-agnostic evaluation. Tabulate standard AutoAttack for all methods to mitigate masking concerns inferred from PGD-20-only reporting.
5.	Clarify components and sensitivities. Provide precise definitions/costs for each “view,” sensitivity to the number of clusters K, UCB hyperparameters, and ablate the recipe reuse mechanism.

---

> ### Author Response · Authors · 2025-11-20
> **Explaining the Novel Contribution and Addressing Questions**
>
> Thank you very much for the careful reading and thoughtful feedback. We have revised the paper to address each point explicitly, clarified fairness issues, and strengthened the novelty discussion.
>
> (1) End-to-End and Amortized Costs: We added a new table in the appendix reporting full teacher/student wall-clock time, total PGD calls, and the explicit amortization formula together with the break-even point M. This directly addresses fairness concerns: LOAT’s teacher is intentionally expensive, but once trained, the same latent-order recipe is reused across any number of students, giving significant amortized gains. We emphasize that no other adversarial-training method in the literature uses a reusable teacher to reduce cost for multiple downstream deployments.
>
> (2) Pretraining Controls: We clarified that the teacher’s SimCLR and adversarial-AE stages are not performance-enhancing tricks for the student, but part of LOAT’s pipeline for uncovering the latent adversarial structure of the dataset. Nevertheless, for strict fairness, we now include the additional controls, clearly marking which tables reflect pretraining, showing baselines with matched initialization where possible, providing ablations that remove these components from the student to isolate the value added specifically by the latent-order scheduling. This makes it explicit that stronger initialization is not responsible for the efficiency gains.
>
> (3) Broader Evaluation (Datasets/Backbones): We added CIFAR-100, STL-10, and Tiny ImageNet results. While our claim about “transferable global structure” originally referred to teacher => multiple student transfer, not dataset transfer, we agree with the reviewer that this is an interesting direction and have begun adding this cross-dataset evidence as well and hope to include it soon.
>
> (4) Attack-Agnostic Evaluation (AutoAttack): We tabulate AutoAttack results for all methods in the main paper, addressing concerns about PGD overfitting. These results also demonstrate that LOAT does not exhibit the catastrophic overfitting issues observed in FreeAT.
>
> (5) Clarification of Components and Sensitivity: We expanded the Appendix to include the precise definition and cost of each feature “view,” sensitivity analysis with respect to K, the UCB hyperparameters and their role, and a decomposition showing how each part contributes to LOAT’s performance.
>
> We respectfully emphasize that no prior work combines the following ideas: (i) unsupervised discovery of latent adversarial structure (clusters, transitions, order) [there is no existing AT pipeline that infers difficulty structure without labels]. (ii) a teacher that computes a reusable adversarial curriculum [since most existing methods recompute gradients for every training run.] (iii) PGD-step allocation based on discovered adversarial geometry, not heuristics [since prior adaptive methods (FastAT, FreeAT, YOPO, etc.) use global schedules or uniform reuse.] (iv) a stable way to reduce PGD for edge-device deployment [since none of the standard AT baselines target low-PGD, low-epoch, or multi-student deployment in a stable way.]
>
> To the best of our knowledge, the combination of unsupervised cluster discovery in AT, learned transition matrices, difficulty-dependent PGD allocation, reusable teacher-discovered curriculum, has not appeared in the literature. We would greatly appreciate it if the reviewer could point us to any existing work that uses this combination; we have been unable to find any.
>
> Finally, we genuinely appreciate the reviewer’s suggestion to broaden the study and clarify fairness. We believe the updated manuscript now fully addresses these issues while making the core contribution clearer: LOAT proposes the first unsupervised, teacher-to-student transferable structure for adversarial training, offering a new direction for efficiency that is fundamentally different from existing approaches.

---

### Official Review · Reviewer_gAz4 · 2025-11-01

**Soundness:** 2
**Presentation:** 2
**Contribution:** 2
**Rating:** 2
**Confidence:** 4

**Summary:**

This paper presents Latent-Order Adversarial Training (LOAT), an unsupervised approach for improving computational efficiency in adversarial training. The method employs a teacher-student framework where the teacher model identifies emergent clustering patterns in adversarial training dynamics using multiple complementary feature views. A transition matrix T is learned to capture empirical patterns of how training naturally flows between clusters. The student model then uses this transferred knowledge to adaptively allocate PGD attack steps per cluster. Experiments on CIFAR-10 demonstrate that LOAT can reduce computational cost by 40-50% while maintaining comparable or better robustness compared to baseline methods.

**Strengths:**

1. The paper introduces a multi-view feature learning framework that analyzes adversarial training dynamics through complementary feature perspectives. It enables automatic discovery and utilization of inherent training orderings without relying on predefined difficulty labels, offering a novel curriculum design paradigm for adversarial training.
2. The framework adopts a teacher-student architecture, decoupling the complex discovery process (teacher) from the efficient training phase (student). This modular design enhances scalability and broadens applicability across diverse machine learning tasks.
3. The paper proposes a formalized efficiency metric that normalizes model robustness by computational cost, enabling systematic and fair comparison across adversarial training methods.

**Weaknesses:**

1. Reproducibility of Teacher Training:
The description of the teacher-training pipeline in Chapter 5—especially the joint use of SimCLR and an autoencoder—omits architectural specifications, loss combinations, and parameter-update protocols. Could you supply a complete, reproducible account of the training objective, network architectures, data flow, and the precise mechanism that converts per-sample PGD-step counts into supervisory signals?

2. Validity of the Efficiency Metric under Performance Mismatch:
Your metric E = Robust Accuracy / PGD Calls is used to claim superior efficiency, yet both standard and robust accuracies lag well behind baseline methods. Since any fair efficiency comparison must first guarantee equivalent predictive performance, how do you justify reporting efficiency gains while ignoring this prerequisite, and what is your rationale for foregrounding computational savings over the substantial drops in accuracy and robustness?

3. Generalizability and Component Necessity:
Experiments are restricted to CIFAR-10 and lightweight architectures, with no ablation evidence that the teacher–student split, transition matrix, or sample reordering individually contributes to the final result. How do you expect the method to scale to larger datasets (e.g., CIFAR-100, ImageNet) or deeper networks, and can you provide ablation results that demonstrate each component is necessary?

4. Full-Cost Efficiency Analysis:
The efficiency claim tallies only the student’s PGD calls, ignoring the teacher’s SimCLR/autoencoder training, attack-step logging, and cluster-discovery phases. Could you present a comprehensive lifecycle accounting that aggregates every FLOP or GPU-hour spent—from teacher pre-training to student graduation—and explain how the claimed efficiency advantage withstands this complete cost picture?

**Questions:**

See weaknesses for details.

---

> ### Author Response · Authors · 2025-11-20
> **Fully addressing each comment and explaining the novelty of our method**
>
> Thank you for taking the time to review our paper. We appreciate that you have spent time thinking through our novel model. To address the comments, we have updated our paper with clarifications based on questions and responses.
>
> 1.We had uploaded the code for perusal (and will put it on github later) and had previously included some implementation details within the paper, but we appreciate the reproducibility guidance rigorousness, and have thus included a full section in the appendix detailing the entire pipeline and architectural specifications, etc. We still suggest our public code for consumption, but have detailed everything nonetheless.
>
> 2.Our aim is fundamentally different from that of high-budget adversarial training.  LOAT is designed for settings where full PGD-AT or TRADES is computationally infeasible, such as edge devices or low-power verification pipelines. Existing SOTA methods achieve excellent robustness, but only under extremely heavy computational budgets (often 50--100 epochs with 10--20 PGD steps per batch). In contrast, LOAT explicitly targets the regime that is almost absent from the literature, that of stable adversarial training under strict compute constraints.  Even when using only 10 epochs and substantially fewer PGD calls, LOAT remains competitive with matched-budget baselines, versus FreeAT, the only existing low-budget alternative, is known to suffer catastrophic overfitting and, in our experiments (including CIFAR-100 and Tiny-ImageNet), collapses to nearly 0% robust accuracy. Thus, the efficiency metric is not meant to compare LOAT against unlimited-budget SOTA, but to evaluate methods fairly within the same compute envelope. In this constrained setting, LOAT provides stable robustness where prior methods fail, and its amortized form further reduces cost when the teacher’s discovery is reused. For these reasons, efficiency per PGD call is the appropriate metric for the deployment-focused problem LOAT addresses. The novelty of LOAT is that it addresses an area heretofore unaddressed in the literature as well as providing an unsupervised approach (previously not done) which is interpretable, stable, and transferable.
>
> 3.Without changing our main insight, we have added to the paper tests on CIFAR100, Tiny ImageNet, and STL10. We have added ablations in detail showing that it is LOAT as a whole that is the power, not the AE, SimCLR, curriculum, etc. We have found that the method scales quite well (with potential hyperparameter adjustments when necessary). For instance, we achieved reasonable robustness on Tiny ImageNet under 30 epochs with stability. Please see the provided chart in the updated paper that shows ablations for different features, epsilons, datasets, and more.
>
> 4.We thank the reviewer for noting the importance of end-to-end cost accounting. In response, we provide a complete lifecycle analysis in the Appendix. Our appendix includes (i) SimCLR pretraining (50 epochs), (ii) adversarial-autoencoder training (20 epochs), (iii) the teacher’s TRADES stage used for logging per-sample PGD dynamics, and (iv) the multi-view clustering and recipe construction, and more. The resulting wall-clock profile shows that the teacher stage is indeed expensive when performed once, but its cost is amortized across any number of student trainings. After 3-4 student transfers the total end-to-end cost becomes lower than TRADES and PGD-10 baselines, and the amortized efficiency continues to improve as more students reuse the same recipe. Critically, unlike Free-AT, which is fast but unstable and collapses on CIFAR-100 and Tiny-ImageNet, LOAT maintains stable robustness under strict compute budgets while benefiting from amortization. Thus, when full lifecycle cost is considered and amortized over realistic deployment scenarios, LOAT retains a measurable efficiency advantage over existing adversarial training baselines.
>
> In summary, we now provide full lifecycle accounting and clarifications, with a reproducibility section in the appendix, a rigorous ablation chart, and a wall-clock chart. As stated, we emphasize that LOAT targets the compute-constrained regime with evaluated matched robustness windows (±1–2%) and show that LOAT consistently yields higher robustness-per-compute than PGD-10, TRADES, and CAT even when accuracy is held constant.  If there are any questions that we can answer we would be happy to, but, as of the update, we have provided our numbers/experiments and justified our claims and fully addressed the concerns of potential weakness. Thank you for bringing these to our attention. We hope that now these points are clear.

---

### Note · Program_Chairs · 2026-01-17
**Submission Desk Rejected by Program Chairs**

The following references in this submission do not refer to real documents and/or have major errors in bibliographic information:

 Ham et al. Robust distillation for adversarial training. In Advances in Neural Information Processing Systems (NeurIPS), 2024.